**Perspective**

# Using computational models of learning to advance cognitive behavioral therapy
Isabel M. Berwian [1] ✉, Peter Hitchcock[2], Sashank Pisupati[1,3], Gila Schoen[4] & Yael Niv [1]

Many psychotherapy interventions have a large evidence base and can help a substantial number of people with symptoms of mental health conditions. However, we still have little understanding of why treatments work. Early advances in psychotherapy, such as the development of exposure therapy, built on theoretical and experimental evidence from Pavlovian and instrumental conditioning. More generally, all psychotherapy achieves change through learning. The past 25 years have seen substantial developments in computational models of learning, with increased computational precision and a focus on multiple learning mechanisms and their interaction. Now might be a good time to formalize psychotherapy interventions as computational models of learning to improve our understanding of mechanisms of change in psychotherapy. To advance research and help bring together a new joint field of theory-driven computational psychotherapy, we first review literature on cognitive behavioral therapy (exposure therapy and cognitive restructuring) and introduce computational models of reinforcement learning and representation learning. We then suggest a mapping of these learning algorithms on change processes presumably underlying the effects of exposure therapy and cognitive restructuring. Finally, we outline how the understanding of interventions through the lens of learning algorithms can inform intervention research.

In recent years, the field of *Computational Psychiatry* has focused on illuminating mechanisms underlying mental health conditions and response to pharmacological and neurobiological treatments using computational models of behavior, decision-making, and biophysiology[1-3]. A family of algorithms often applied in computational psychiatry originate from reinforcement learning (RL) and prescribe a set of rules for trial-and-error learning from reward and punishment. Bayesian inference, also widely used in computational psychiatry, accounts for learning through combining prior beliefs and new evidence. While these computational models of learning have been applied to studying disorders ranging from depression and anxiety to schizophrenia, they have been rarely applied to studying psychotherapy as a means of treating such disorders (but see refs. 4,5). As psychotherapy embraces many interventions that are directly based on learning theory, it readily lends itself to be studied and quantified through the lens of computational models of learning. Expanding on Niv et al.[6], this paper aims to build a bridge between learning algorithms and psychotherapy research and point out ways in which computational models of learning, together with well-designed behavioral tasks, could enhance psychotherapy research. Within psychotherapy, we will focus on (second generation) *cognitive behavioral therapy* (CBT) due to its straightforward mapping to learning algorithms and its wide clinical use. However, the

principles of our approach can be similarly applied to elucidate mechanisms underlying psychotherapy interventions from other schools of thought (e.g., psychodynamic therapy and interpersonal therapy).

CBT is a problem-focused collaborative form of psychotherapy that aims to change maladaptive behavior and thought processes[7,8]. One core assumption of the behavioral part of CBT is that people expect (explicitly or implicitly) that specific situations lead to dangerous outcomes when they are, in fact, safe. "Exposure therapy" therefore attempts to change people' maladaptive responses and behaviors through direct new experiences: people are exposed to feared situations to change previously learned associations between the situation and danger, or train new competing associations between the situation and safety. Such learning can help reduce maladaptive emotional and instrumental responses, such as a misplaced autonomic fear response and/or avoidance behavior.

A core premise of the cognitive part of CBT is that external events do not cause us to feel and do things; instead, our cognitions offer a subjective interpretation of events that, in turn, causes feelings and actions. This interpretation is often automatic and implicit, building on a lifetime of previous experiences or modeling by others, and is not necessarily a voluntary or conscious process. The profound implication is that by changing our interpretations, we can avoid responding maladaptively

[1]Princeton Neuroscience Institute & Psychology Department, Princeton University, Princeton, NJ, USA. [2]Emory University Psychology Department, Emory University, Atlanta, GA, USA. [3]Atla AI Ltd, London, UK. [4]Geha Mental Health Center, Petah Tikva, Israel. ✉e-mail: iberwian@princeton.edu

(for examples, see below). "Cognitive restructuring" in CBT addresses these interpretations by challenging thoughts, exposing their exaggerated or distorted nature, and listing and practicing alternative interpretations. This, too, can help reduce or change the emotional response and enable alternative behavioral responses.

Below we briefly explain, for each of these interventions (exposure therapy and cognitive restructuring), what they entail, how they are delivered in practice by a therapist, and their underlying theoretical assumptions (and empirical evidence). We then introduce relevant terminology from reinforcement learning, representation learning and Bayesian inference. Finally, we map each intervention onto specific learning algorithms and carve out predictions regarding enduring treatment response and relapse.

Before diving into the details of each intervention, we would like to point out that these interventions are usually embedded in a therapy framework that includes a diagnostic assessment, the building of an alliance between the client and the therapist, psychoeducation about the mental health condition and planned approach, 5−20 sessions of the primary intervention(s), and relapse prevention. It is important to keep in mind the influences of the additional components on therapy outcome, and future work should integrate them more precisely into the outlined framework (see Discussion). We would also like to emphasize that we do not suggest implementing the modified interventions we outline below in the clinic at this point, rather, we propose these as hypotheses that first must be tested experimentally and clinically. We hope that testing interventions that are informed by learning systems in the brain can help elucidate their mechanisms of action, and suggest refinements to existing treatments.

Furthermore, we note that many verbal learning theories have been put forward in psychology, and these form the basis of psychotherapy research (for example, theories of context learning[9,10]). Additionally, empirical and neural evidence, e.g., for generalization learning[11–13], informs psychotherapy research (see ref. 14 for an example). Computational models are not alternatives to these learning theories, but rather they are mathematical formalizations of them. These models make implicit assumptions explicit, identify gaps in traditional learning theories, and mathematically define how and when learning occurs. Thus, they allow for precise and quantitative prediction of learning behavior over time. As such, they can be used to test, quantitatively and qualitatively, if theories can indeed account for observed learning behavior and which theory does so best. This can help identify explanatory gaps and prompt elaboration of the theory (for example, see ref. 15). Reviewing all traditional learning theories is beyond the scope of this paper, however, we will point the interested reader to relevant traditional theories throughout.

## Exposure therapy

Exposure therapy is the oldest part of CBT. It is a widespread and effective treatment for anxiety disorders[16–19] and especially effective for treating post-traumatic stress disorder[20], specific phobias[21,22], obsessive-compulsive disorder[23] and other mental health conditions.

In exposure therapy, people repeatedly encounter real, simulated, or imagined feared stimuli within safe environments. According to theories of exposure therapy, they then learn to remain in the presence of the feared stimulus or situation rather than to escape it, and experience that the feared outcome does not occur. Subsequently, their fear of the stimulus decreases and with it the tendency to avoid the stimulus.

Different strategies of exposure therapy include directly facing the feared stimulus or situation (in vivo exposure), vividly imagining the feared stimulus or situation (imaginal exposure), using virtual reality to show the feared stimulus or situation (virtual reality exposure), or purposefully bringing about a feared physical sensation (interoceptive exposure). In preparation for exposure therapy, the therapist and client construct a list organizing fearful stimuli and situations according to the strength of the client's fear ("fear hierarchy"). For example, for a client struggling with social anxiety, at the bottom of the hierarchy (less fear) may be situations such as meeting with 2−3 friends, whereas at the top may be giving a public talk. In gradual exposure, exposures start with mildly or moderately feared

situations and subsequently move up the hierarchy. In contrast, in flooding, the client is immediately exposed to the most fearful stimulus or situation[24].

Exposure therapy is based on the assumption that avoidance of fearful stimuli and situations maintains fear and anxiety[25,26]. The idea is that fear was learned when a harmful outcome occurred. Due to subsequent avoidance of the fearful situation or stimulus we do not experience that it rarely (if at all) leads to the aversive outcome, and thus we miss opportunities to correct our exaggerated expectation of the fearful outcome and decrease our fear.

Foa and Kozak[27] suggested three factors that indicate effective exposure therapy: (1) The person is in a fearful state in response to the stimulus, which Foa and Kozak termed "fear activation." In their theory, fear activation indicates that the fear *memory* has been accessed and can be integrated with new information to achieve emotional change and is a prerequisite for learning to occur. (2) The person's physiological response to the stimulus decreases *within* the exposure. That is, the exposure exercise must be long enough to experience a reduction in subjective distress (note that this factor is no longer considered required for effective exposure therapy due to a lack of empirical evidence[28]). (3) The person's physiological response to the stimulus decreases *between* sessions as a sign of habituation due to repeated exposure. The latter two are presumed to occur through the natural decrease of the body's autonomous stress response through habituation to a non-fearful situation – even absent mitigating actions, stress responses tend to decrease after 15−20 min.

Craske et al.[29,30] suggested an alternative tactic for improving exposure therapy that does not rely on habituation but on "inhibitory learning". Rather than modifying the original fear association, inhibitory learning focuses on forming and strengthening new inhibitory associations between the stimulus and safety (therefore, the new association is inhibitory to the fear association). Specifically, Craske and colleagues suggest that maximizing the *difference* between the expected (aversive) outcome and the actual outcome (expectancy violation) during exposure will increase inhibitory learning. Based on empirical evidence of the role of inhibitory retrieval in extinction learning, they developed a range of techniques to improve exposure therapy outcomes[14]. These techniques include what they call foundational strategies that enhance expectancy violation (e.g., attention to the feared stimulus, removal of safety signals), advanced strategies (deepened extinction, i.e., extinction of multiple feared stimuli, first separately and then combined, and occasional pairing of the feared stimulus with the aversive outcome) and strategies to enhance the generalization of extinction learning (using retrieval cues as reminders of the extinction experience and practicing extinction in multiple contexts or for a variety of stimuli similar to the feared one).

While exposure therapy is successful in reducing fear for many clients, fear often returns after some time[31]. Indeed, the prevention of relapses might remain one of the biggest challenges for exposure therapies. This challenge provides an opportunity for computational psychotherapy to improve therapy design such as to reduce return of fear, as we detail below.

## Introducing model-free reinforcement learning

Reinforcement learning (RL) arose at the interface of computer science and psychology as a computational theory of animal learning, specifically Pavlovian and instrumental conditioning[32]. Before introducing RL, we will therefore introduce Pavlovian conditioning, which is also important for understanding exposure therapy. In Pavlovian ("classical") conditioning, a contingency between two events—one motivationally neutral and one motivationally relevant—is experienced repeatedly (e.g., a light – a "conditional stimulus," or CS, followed by a shock – an "unconditional stimulus" or US). As a result, through learning, the CS comes to *predict* the occurrence of the US. This prediction is evidenced (and can be measured) by behavioral responses (e.g., quickening heart rate – also called "conditioned responses") that automatically accompany said prediction. This type of learning is ubiquitous and can occur when a CS predicts the occurrence of a US (excitatory conditioning) or the absence of a previously occurring US (inhibitory conditioning). When a CS that previously predicted a US is

repeatedly presented without the US, the conditioned responses decrease. This process is termed *extinction learning*. Importantly, although Pavlovian learning may not even enter awareness, Pavlovian conditioned responses (which include emotions, increased heart rate, and sweating) are automatic and very hard to override. In everyday life, emotional responses (e.g., anxiety to the sight of a co-worker approaching your desk at noon) are often conditioned in this way (e.g., due to experiencing an aversive lunch interaction with this co-worker in the past). Here, the co-worker serves as the CS, and the heightened arousal and anxiety are a conditioned response.

Prediction learning is at the heart of RL theory. In RL models of Pavlovian conditioning, a stimulus or "state" $S$ (which refers to the constellation of stimuli co-occurring at a specific time, such as background, location, not an internal bodily or mental state) acquires a value $V(S)$ that reflects the subjective scalar sum of the motivationally relevant USs it predicts. This occurs through repeated experience with environmental contingencies and learning through prediction errors. For example, the first time the light is seen, the value of the "light on" state may be zero as it does not inherently predict anything of motivational value. When the light is followed by a shock, the shock's "reward value" $R$ (negative for aversive USs such as a shock; positive for appetitive USs such as food) is compared to the prior expectation $V_{old}(S)$ to compute a prediction error: $PE = R - V_{old}(S)$. The new value of the state is then updated based on the prediction error: $V_{new}(S) = V_{old}(S) + \alpha \cdot PE$, where $\alpha$ is a "learning rate" parameter between 0 and 1. As the value will now be larger than 0, the light will now partly predict the shock on the next trial, leading to a smaller prediction error (and a smaller value update). The learning rule will update the value until the prediction error is zero (that is, until the prediction is correct). The earliest version of this model was proposed by Rescorla and Wagner[33] to explain how an association between a CS and a US is acquired.

This model has one parameter, $\alpha$, that determines the extent to which new information about USs is incorporated into future predictions, at the expense of old knowledge. $\alpha$ can differ between individual learners or for different situations, explaining differences in the rate of prediction learning. Note that this learning model is an example of "model-free RL" because it does not require learning a "world model" – the actual environmental contingencies. Even without explicitly tracking the probability that the US occurs, or the size of its reward, correct predictive values can be learned through trial and error, from prediction errors.

Model-free learning operates on states $S$ that represent meaningful or relevant configurations of stimuli. Although in the theory states are assumed to be known, in reality, they are not pre-defined for the learner. States must also be learned by each individual through what we will call "representation learning" – discovering what configurations are relevant in the current situation. For instance, one might learn that each of a specific group of co-workers approaching around noon represent a state with aversive value (as they tend to make offensive remarks about other colleagues at lunch), while other co-workers represent neutral or even positive-valued states. Even in simple scenarios as in the light-shock conditioning situation described above, the light may be relevant for predicting shocks, whereas environmental odors and one's location in space may not be relevant. However, in other scenarios, locations may be relevant.

"Latent-cause inference" is a recent computational framework that addresses the problem of learning a high level representation for an RL task[34]. In this theory, learners infer "latent causes" – groupings of events into clusters, or contexts, in which certain contingencies between events can be expected. Latent causes can function as the states from traditional RL. For instance, in the example above, the learner can infer a latent cause $C_1$ in which both light and shock appear with high probability. When the environment changes considerably (e.g., the light that once reliably predicted a shock is no longer followed by a shock, as in extinction), traditional RL theories of conditioning would suggest new prediction errors and updating of $V(S)$ to match the new reality. However, according to latent-cause inference, the learner may instead infer that there is now a new latent cause $C_2$ in which lights are likely, but shocks are not. Importantly, in this theory, separate values are learned for each latent cause, so the newly learned

value of the light (now zero) does not interfere with the high predictive value of the light learned in $C_1$.

Different factors might promote updating of an old latent cause or the inference and updating of a new one. According to the theory, in fact, both options co-exist. For example, on a specific trial, $C_2$ may be inferred to be in effect with some probability (say, 0.2) and $C_1$ assigned the remaining probability (0.8). Given that the agent does not know with certainty which latent cause is in effect, the likelihood of a shock occurring in both causes will be updated with a learning rate proportional to the probability of that latent cause. This allows the learner to apportion learning to all likely active latent causes. Nevertheless, for simplicity, we will refer throughout the remaining text to "creating a new latent cause" when suggesting that a new latent cause is inferred with a high probability for the first time, and to "updating the old latent cause" to indicate that the old latent cause is inferred with high probability.

## Using computational theories to explain exposure therapy

There is considerable evidence for both RL and latent-cause inference[32,35,36]. We now use these theories to explain treatment outcomes in exposure therapy.

### Explaining exposure therapy in terms of the latent-cause framework

Pavlovian conditioning and extinction learning were key to the development of exposure therapy (e.g., refs. 25,26,29,37). One key challenge is to prevent relapse, or the resurgence of symptoms after successful exposure therapy. In psychotherapy research, relapse has often been explained by inhibitory learning, defined in the psychotherapy literature as the formation of a new association between the CS and no US through therapy. Note that this is different from inhibitory conditioning or conditioned inhibition, where the presence of a CS predicts the absence of an otherwise expected US[38]. In successful inhibitory learning, the new CS-no US association competes with the original CS-US association successfully, leading to reduction of fear. However, if the competitive edge of the CS-US association is stronger at a later time point, the fear returns and relapse can occur[9,29]. In the lab, such return of fear after extinction training has often been observed after the passage of time (termed 'spontaneous recovery'), after a context switch ('renewal') or after re-exposure to the US alone ('reinstatement'). Several theories have been proposed to explain spontaneous recovery (see ref. 39 for a review) and context learning[10] is the most prominent theory to account for renewal and reinstatement. RL theory, particularly its extension to latent-cause inference, provides a computationally formalized explanation for the processes underlying extinction and the return of fear. This formalization of previous theories, such as[10] context learning, within a mathematical framework that can be quantitatively fit to data, has the advantage that it allows fitting parameters of the model to the behavior of participants and in this way quantifying and studying individual differences across people. As such, RL and latent-cause formalisms can suggest additional routes to preventing relapse and make testable predictions about indicators of what would be the best route for each individual, as we will detail below (see ref. 15 for an example).

According to the theory, a person exposed to a feared stimulus (CS) in a safe scenario (that is, in the absence of the feared US) could either update the value of the feared stimulus according to the prediction error, or infer the existence of a different, "safe" latent cause, and learn a separate value for the stimulus when that latent cause is active (Fig. 1). The two types of updates will have different behavioral signatures and long-term consequences. Updating the old value of the stimulus, e.g., through reconsolidation of the retrieved memory of the stimulus[40,41], might require several rounds of exposure, and therefore will lead to a slow (but long-lasting) reduction of fear. In contrast, creating a new latent cause can lead to rapid reduction of fear in response to the stimulus, given that the new latent cause was never associated with fearful events (so their probability under that cause is zero or close to zero).

**Fig. 1 | Latent-cause inference.** After observing a CS (e.g., a light) followed by a US (e.g., a shock), the CS comes to predict the US. The prediction is mediated by a latent cause $C_1$ that jointly "emits" both CS and US. In exposure therapy, when the CS is observed but not followed by a US, the prediction can be either weakened through prediction-error driven learning (Option 1: dashed line signifying lower probability of the latent cause emitting the US, as per the new learning) or a new latent cause $C_2$ is created, which only emits the CS (Option 2). CS = conditioned stimulus, US = unconditioned stimulus.

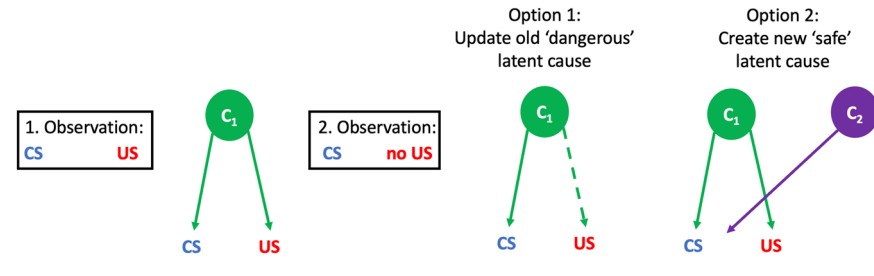

However, creating a new latent cause comes with a risk of return of fear if the old, unchanged latent cause is inferred to be active again.

Hence, according to the latent-cause framework, relapse can occur when a new 'safe' latent cause is created during exposure, but at a later timepoint, the old 'dangerous' latent cause is inferred. The framework thus predicts two different (and opposing) routes to preventing relapse. One route targets the original 'dangerous' latent cause, reactivating and updating it. This should cause long-lasting change as once the 'dangerous' latent cause is updated, it no longer predicts danger and relapse is, at least theoretically, impossible. A second route is to create a new 'safe' latent cause and train this latent cause such that it is preferentially inferred and generalizes as widely as possible. In a Bayesian framework, this inference depends on both the probability of each cause being active independent of any observations (that is, the *prior* belief in each of the causes) and the probability of the observed stimuli given each latent cause (the *likelihood* of the observations, typically determined by their similarity to stimuli previously linked to the latent cause). Thus, to strengthen generalization, one could increase the prior probability of the safe latent cause through many learning experiences in different contexts and situations[14].

**Conditions for learning due to exposure**

If one does not expect a negative outcome, exposure therapy will not work, as there will be no violation of expectations and no new learning (updating of a latent cause and/or inference of a new latent cause). This could happen, for example, if a person believes they are in a safe situation due to a safety cue (e.g., the therapist's office). Signatures that one is expecting a negative outcome include behavioral or physiological fear responses and verbal report of emotional arousal, also termed 'emotional engagement'. Many types of psychotherapy emphasize the importance of clients' emotional engagement during therapy (e.g., ref. 27). We propose that negative outcome expectations result from inferring that the 'dangerous' latent cause is active. In the latent-cause framework, this inference is based on *prior* (learned) beliefs, formalizing the effect of 'core beliefs' (a term from psychotherapy) on predictions and expectations. For example, consider a person who was bullied and humiliated by others. The formation of a latent cause that ties people in general (CS in Fig. 1) to personal harm (US in Fig. 1) embodies a belief that 'other people are dangerous.' The person likely also had positive experiences with other people and had formed the other belief 'some people can be nice' (in a separate latent cause). If the dangerous latent cause is sufficiently "strong" so that it is predominantly inferred when meeting a new person (that is, even before observing the unfamiliar person's behavior, there is an expectation that they might cause harm), the person in question might feel afraid, show anxious behavior and avoid new people. The dangerous latent cause would then be considered a 'core belief' as it is applied widely as a prior even without observing situation-specific evidence.

Latent-cause inference occurs whenever new information is available, because any new information – whether motivationally neutral or aversive/appetitive – can help refine inference of the current latent cause. If the person above meets someone new, they might show anxious behavior and avoidance due to inferring a 'dangerous' latent cause ($C_1$ in Fig. 1). When they experience no harm, they might infer that, in fact, a 'safe' latent cause

($C_2$) was active, and update that latent cause, leaving the 'dangerous' latent cause unchanged. Therefore, initial fear behavior can be triggered by one latent cause inferred due to a CS, whereas learning upon receiving the US may be applied to a different latent cause. This distinction between latent cause inference before and after outcome observation can account for the sometimes apparent disconnect between what drives actions and what is learned. While psychotherapy interventions focus on ensuring that the dangerous latent cause is initially activated (the first prerequisite for learning according to Foa & Kozak[27], and the expectation of a negative outcome according to Craske et al.[29]), they seldom ensure that it is activated when learning takes place after outcome observation. Thus "fear activation" may occur when the previously bullied person meets a new person in an exposure exercise. However, if the person attributes the friendliness of the new person to the location of the meeting (that is, they infer a latent cause $C_2$ for meetings around the clinic), fear may not decrease for future meetings elsewhere.

**Avoiding the creation of a new latent cause**

According to this theory, relapse can be avoided entirely if the 'dangerous' latent cause is updated rather than a separate, 'safe', latent cause. Thus, it is important to understand under which circumstances new latent causes are inferred or how separate latent causes can be combined into one. One important determinant of latent cause inference is the difference between observed events and those predicted under each of the already-learned latent causes. An outcome that is not predicted under existing latent causes is referred to as a prediction error in RL and expectancy violation in the psychotherapy literature. Empirical evidence indicates that the magnitude of this expectancy violation influences whether a new latent cause is created[42–44]. In particular, large expectancy violations mean that the likelihood of the observation is small under known latent causes, which favors inferring that a new latent cause is predominantly active. More moderate expectancy violations might tilt the balance towards a new latent cause to a lesser extent, and instead lead to updating of old latent causes. Thus, exposures that result in smaller expectation violations may promote fear reduction through updating of existing "dangerous" latent causes, while the inhibitory learning approach that maximizes expectation violation[29] promotes fear reduction through creation of a new "safe" latent cause. In the latter, care should be taken to ensure the new latent cause generalizes to many situations and is maintained over time, to prevent relapse due to the old "dangerous" latent cause resurfacing.

Beliefs about the way latent causes work (that is, their *priors* for new latent causes) may also impact individual tendencies to create new latent causes. A person who holds the belief that the world is mostly deterministic, that is, that latent causes emit the same observations every time they are active, is more likely to infer a larger number of latent causes to explain varying observations. For instance, someone who (implicitly) assumes that people are either good all the time or bad all the time ("black-and-white thinking") will require more latent causes to account for the sometimes good and sometimes bad behavior of their partner (e.g., a "good days" latent cause and a "bad days" latent cause), whereas someone whose prior beliefs are that no one is perfect and everyone has good and bad sides will more readily infer

one latent cause for all their partner's behaviors. The former latent-cause inference might underlie behavior that resembles that of people with Borderline Personality Disorder[45]. At the other extreme, someone who holds the belief that the world is very stochastic and not predictable might have only one latent cause for all people – if they were harmed by others early on, they might then overgeneralize their expectation of harm to all other people.

Paradoxically, assuming high (but not complete) stability of latent cause across time can also result in inference of new latent causes. In particular, simulations of the latent-cause inference model show that people with a stability prior are more likely to "stick" to the new latent cause that is always inferred (with some small probability) when contradicting information is encountered. Thus, they too will tend to not incorporate the change into previous latent causes[46,47]. Both deterministic and stability beliefs can therefore lead to rapid reduction of fear during exposure therapy, as the lack of negative outcomes would promote a new latent cause (due to determinism), which would then be stable throughout therapy (due to beliefs that the world shows stable relationships between events). But these beliefs also protect the old dangerous latent cause, which can resurface in a later phase. Alternatively, selectively maintenance of adverse experiences can also explain spontaneous recovery, as over time the strength of aversive memories increases relative to memory for neutral events such as the extinction experience[15].

Rigid beliefs, i.e., believing the world is either very deterministic or very stochastic, that it never changes, or it changes all the time, are represented by extreme parameters in the model. In the face of such rigid prior beliefs, new information about the world plays a minor role in the inference process. In contrast, flexible beliefs might allow for adaptation to different circumstances and contexts and integration of new information as appropriate. The latter might lead to better outcomes in therapy, while the former might underlie various mental health issues such as borderline personality disorder[45], anxiety disorders[48] and obsessive-compulsive disorder[49].

Note that these prior beliefs about the world in general are another instance of core beliefs (Is the world reliable or not? Can previous experience be trusted to be useful in the future? Etc.), which are most likely shaped by early life experiences. That is, in theory, prior beliefs about latent causes can also be learned from experience, and one can imagine this learning is more pronounced in childhood, with priors becoming more entrenched later in life.

### Refining treatment

The latent-cause framework explains why behavioral measures such as speed of fear reduction cannot reliably predict response to exposure therapy or long-term relapse (as reviewed in ref. [30]). This is because fear reduction might result from the creation of a new latent cause or fast updating of an existing cause, two mechanisms that can lead to similar behavior. Instead, we suggest that quantifying individuals' priors over latent causes, tendency to create new latent causes and selective maintenance of negative events can help determine for whom exposure therapy will be more effective in the long run, and what specific form of exposure therapy (e.g., with small or large expectation violations) should be applied. To do this, one can construct learning tasks to which models of latent-cause inference can be fit to determine individual parameters of the inference process (see ref. [15]). Specifically, we predict that people who more readily create new latent causes and have a strong tendence to selectively remember emotionally relevant events will have a higher risk of relapse after exposure therapy (see, e.g., refs. [15,50]). In contrast, those who create very few latent causes that generalize across a wide array of experiences will show slow learning during exposure therapy (their fear might even seem resistant to change), but this learning will be more long-lasting.

Poor outcomes of exposure therapy could be mitigated by targeting differences in latent-cause inference *before* exposure therapy. For example, a person's deterministic/dichotomous beliefs about the world can be targeted with cognitive restructuring before exposure. Those who overgeneralize in the interpersonal domain might benefit from the interpersonal discrimination exercise in Cognitive Behavioral Analysis System of Psychotherapy (CBASP)[51], which emphasizes the differences between oneself

and other significant people in one's life and thus can help create new latent causes and new expectations about behavior of others.

We can also combine our knowledge of mechanisms underlying response to exposure therapy with individual differences to design interventions that match clients' learning. For example, if a client has the tendency to update old 'dangerous' latent causes, ensuring that the 'dangerous' latent cause is activated during exposure by creating a situation that closely maps the situation of fear acquisition is important. In contrast, if a client tends to create new latent causes, it would be crucial to enhance the generalization of the new latent causes. This is achieved by increasing their prior probability through, for example, variability of context and stimuli during exposure as suggested by Craske and colleagues[29].

### Cognitive restructuring

Cognitive therapy is based on the assumption that events trigger cognitions, also referred to as automatic thoughts, which trigger negative emotions. Based on this causal sequence[8,52], Beck suggested that correction of distorted automatic thoughts is a crucial step in reducing negative emotions. These corrections are at the heart of *cognitive restructuring*, the core intervention of cognitive therapy.

Initiating the cognitive revolution of psychotherapy, rational emotive therapy[53] (now known as rational emotive behavioral therapy) and Beckian cognitive therapy for depression[7,8,52,54] were based on empirical evidence and clinical observations that clients with depression suffer from negative cognitive distortions about their experiences and predictions of the future. Although cognitive restructuring was developed as a treatment for depression, it can be also used for treating many mental health conditions that show signs of distorted cognitions, such as anxiety, obsessive-compulsive disorder and psychosis (for example[55,56]).

Cognitive restructuring includes four steps: (1) identifying the automatic thought, (2) identifying in what ways the automatic thought is distorted, (3) challenging the distortions (e.g., by listing evidence for and against a specific interpretation to showcase that the thought is likely an incorrect interpretation of the event), and (4) rebuttal of the thought (for instance, by challenging the client to defend the new interpretation). These steps are often supported by handouts. Clients are asked to write down the triggering situation, the automatic thought, and their emotion in different columns. They are then asked to detail the evidence for the thought, the evidence against the thought, suggest at least one alternative thought, and rate their emotion given the alternative thought (see example in Box 1). Automatic thoughts are often specific to a situation but are assumed to be driven by underlying core beliefs about the self that are more general, such as 'I am a failure'. As discussed above, such core beliefs are often formed during childhood and are more resistant to change. Cognitive restructuring aims to address core beliefs as well, either through changes in automatic thoughts or by tackling them directly at an advanced stage of the treatment[57].

This procedure aims to make explicit and change the client's own underlying assumptions and discrepancies between their assumptions and experiences from the real world. To achieve this, therapists employ dialectic techniques such as the Socratic method, with the goal of having clients arrive at the solution or alternative view on their own, while the therapist only guides them through questions.

Cognitive therapy and cognitive restructuring have empirical support[58–60]. However, an early component study found no advantage of enhancing behavioral therapy with cognitive therapy for patients with Major Depressive Disorder[61]. A recent large component network meta-analysis of internet-delivered CBT for people with depressive symptoms also found weaker effects of cognitive restructuring compared to other interventions[62]. As with exposure therapy, it seems that cognitive restructuring works for some, but not for everyone[63]. Here, too, we propose that we might be able to improve interventions and identify who stands to benefit from cognitive restructuring by using our understanding of the change processes involved in cognitive restructuring and their necessary prerequisites. Below, we use RL theory to identify the change processes and hypothesize prerequisites for response to cognitive restructuring.

## Box 1 | Example of cognitive restructuring

A client presents with worries and anxiety related to doing their job, as well as a resulting loss of interest in the job.

**Situation:**
I made a mistake at work.

**Automatic thought:**
When my boss learns about my mistake, they will fire me.

**(Core belief:**
I am terrible at my job and deserve to be fired.)

**Emotional response and mood:**
Afraid, worried, loss of interest in task in job, sad

**Evidence for thought:**
When my last boss found out about mistakes I made, I was punished.

**Evidence against thought:**
My current boss gave me positive feedback in my last meetings.
They offered me a promotion three months ago.
When my colleague reported a mistake, they were not fired.
My boss says they want to be informed about mistakes and the mistakes will have no personal negative consequences.

**Alternative thought:**
My boss might ask me to correct the mistake but will not fire me.

## Model-based decision making

Above, we described model-free RL, where values for states are learned from experience. Another way to estimate the predictive value of a state is to learn a *model of the environment*: how states follow one another given each action (the state "transition structure"), and what states are rewarding or punishing. Armed with this model, the learner can mentally simulate the consequences of different actions (or, in Pavlovian scenarios where actions are irrelevant, the unfolding of events over time) to calculate their value. This alternative algorithm has been termed "model-based learning," because it relies on a model of the environment, in contrast to the "model-free" trial-and-error learning algorithm[64]. Model-free and model-based learning can be seen as formalizations of[65] 'law of effect' and[66] 'cognitive maps', respectively. This distinction also relates to dual-process theories of cognition[67], with model-free learning and decision making associated with the more 'impulsive' System I and model-based decisions related to the more 'deliberative' System 2.

Model-based action selection has been associated with deliberative, so-called goal-directed behavior. This kind of behavior is more flexible than actions based on model-free values (which are considered more habitual), as the model (and therefore, the action values) can incorporate new information without extensive experience. For instance, if you meet a new person and a shared acquaintance tells you this person is trustworthy, you can immediately update your model and interact accordingly, even if your prior had been that people are not to be trusted. However, using cognitive deliberation (e.g., simulation of future outcomes for different courses of action in RL terms) requires mental effort, and there may be limits to how deeply one can search a tree of future options. As such, we often rely on habitual, model-free behavior rather than employ "expensive" model-based decision making. Neural and behavioral evidence indeed shows that both systems operate in the brain in parallel, with one or the other system controlling behavior at any point in time[64]. For example, you may use model-based decision making to plan how to have a conversation with your partner about an issue that has been bothering you, without it devolving to the usual argument. However, as the conversation proceeds and you are confronted with unplanned responses from your partner, you feel attacked, triggering your "danger" response. As a result, your model-free system takes over your emotional and verbal actions.

## Mapping cognitive restructuring onto reinforcement learning

Cognitive restructuring presumably updates the internal world model used to make deliberative, goal-directed decisions[64]. For example, in Box 1, after the cognitive restructuring exercises, the client knows that their expectation that their boss will fire them when they tell them about a small mistake is unwarranted. They can therefore change the probability of ending up in the "fired" state in their internal model to be much lower, which will immediately result in a higher computed value for the action of disclosing the mistake to their boss. This change can occur despite not yet experiencing that situation, and the prediction errors it may entail.

Nevertheless, the client may feel afraid and avoid discussing their mistake with their boss because their model-free valuation of disclosing has not changed. In particular, model-free values are thought to be implicit, not necessarily available to awareness, and not modifiable at will through explicit cognitive processes[64,68,69]. While some people may be able to deploy model-based decisions despite model-free values suggesting these actions are dangerous, others may not be able to override their model-free values as readily. Therefore, for cognitive restructuring to be maximally effective, it is important for clients to translate changes in explicit, model-based beliefs into changes in implicit, model-free values. Cognitive theories of psychotherapy indeed assume a direct influence of changes in cognition on changes in emotions. However, they remain elusive about the underlying mechanisms. We suggest that mental simulation is one mechanism by which cognition (i.e., the model-based system) may influence model-free values. Mentally simulating a situation with the new cognitive information embedded in it (e.g., playing the conversation with the boss in your mind), can offer "pseudo-experiences" from the model-based system to train model-free values[32,70]. Such mental simulation has been shown to affect future decisions in humans[71]. The effect of mental simulation on model-free values could be enhanced by encouraging clients to engage mental imagery so that neural activations more closely mimic real experience[36,72].

Like with exposure therapy, although model-based (explicit) changes in cognition might, at first glance, lead to a rapid improvement, this improvement may not last. Rapid improvement may be due to using model-based decision making to act on the updated model or transient updates in working memory[73], without changes to model-free values. If that is the case, when experiencing stress or when cognitive resources are limited—situations in which decisions rely more on model-free (implicit) values[74–77]—old response patterns that are embodied in the model-free system may resurface. This suggests a possible mechanism of relapse due to stress and underlines the importance of translating model-based (explicit) knowledge to model-free (implicit) values to achieve enduring change. Figure 2 illustrates the proposed model of updating and action selection.

We accordingly hypothesize that encouraging mental simulation and mental imagery could increase the effect of cognitive restructuring by enhancing model-free learning absent direct experience, and that people who more readily translate model-based learning to model-free changes will show a more enduring response to cognitive restructuring. This ability can be potentially assessed experimentally to predict enduring treatment response.

A second type of learning can be (and often is) engaged by cognitive restructuring: acquiring the ability to identify cognitive distortions and apply the questioning and change procedure to new thoughts. This is also termed 'meta-learning'. The relevance of a person's capacity for meta-learning and ways to assess it have been outlined by Reiter et al.[78]. One important aspect of meta-learning is metacognition, the ability to monitor and appraise one's cognitive experiences. People with symptoms of anxiety and depression often show a metacognitive bias, i.e., a reduced confidence in their own performance[79]. Anxious-depressive symptoms have been shown to reduce after internet-delivered CBT. This reduction was accompanied by increased confidence in one's performance (a reduction of the metacognitive bias)[80]. Both meta-learning and

translation of changes in the model-based system to model-free values are relevant for the ability to apply learned cognitive strategies to new thoughts outside or after the end of therapy.

Finally, cognitive restructuring might not only target the model of the world as discussed above, but also latent-cause inference. Changing cognitive beliefs may influence which state is inferred in a specific situation, or how boundaries between states are drawn.

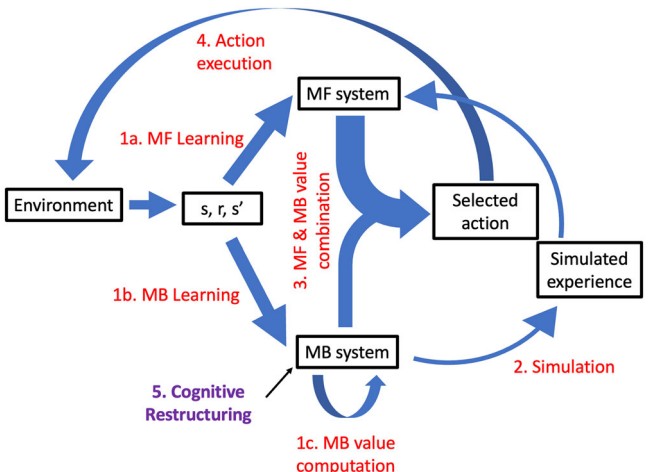

**Fig. 2 | Intuitive depiction of a model with interacting model-free and model-based components.** The environment provides information about the current state (s), the reward (r) received, and the transition to the next state (s'). This information can be used to update state and action values in the model-free (MF) system through computing prediction errors (1a), and to learn reward and transition matrices in the model-based (MB) system (1b). Using the latter, which correspond to a 'model' of the environment, the model-based system can compute state and action values (1c). The model-based system can use this information to train the model-free system by simulating experiences, however, simulations may exert weaker influence than real experiences (2). Actions are selected (3) based on a combination of the model-free system's values and the model-based system's values. In case of stress or time pressure, the model-free system influences action selection more strongly, as indicated by the thicker arrow. The selected action is then executed and influences the environment (4). We propose that one target of cognitive restructuring is changing the reward and/or transition matrix in the 'model' stored in the model-based system (5).

## Discussion

CBT has its roots in decades of experiments on animal learning and corresponding theory. Harnessing ideas from contemporary computational models of learning might help develop even more effective treatment protocols. Towards this goal, here we mapped terminology from computational models of learning onto concepts and interventions applied in psychotherapy. We outlined the relevant formulations of reinforcement learning theory and latent-cause inference, as well as the implementation and development of exposure therapy and cognitive restructuring. Mapping these fields onto each other allowed us to make two proposals. First, we propose that quantification of individual differences in inference about the structure of the world (i.e., latent-cause inference) may be used to predict response to exposure therapy and refine treatment protocols for those who are at risk for relapse. Second, we suggest that the ability to generalize model-based (explicit) knowledge to model-free (implicit) values may be predictive of better response to cognitive restructuring therapy.

### Testing the predictions

We so far provided a theoretical mapping between computational models of learning and psychotherapy research. To test the hypotheses generated with this approach, we need to measure learning tendencies and abilities in individuals. In Fig. 3, we illustrate this approach using a behavioral task that measures learning during extinction training, which emulates exposure therapy. We fit a version of the latent-cause model introduced above to behavior elicited by the task in order to quantify underlying learning tendencies (for details on the technical steps involved in fitting models to behavioral data, see Wilson et al.[81]). Our results suggested that some individuals show spontaneous recovery of fear with time, while other do not and that this difference could be related to differences in selective maintenance of adverse events in the model[15]. This task, and its related computational model allows us to ask new questions, e.g., whether we can predict who will be at higher risk of spontaneous recovery (relapse) of fear after exposure therapy for anxiety, and whether the model's insights into the mechanism of spontaneous recovery can help to modify treatments so as to prevent relapse from occurring.

As shown in the example in Fig. 3, tasks, models, and parameter estimates can be used to address such questions in two ways: First, we can train a machine-learning algorithm to predict relapse using model-derived parameter estimates of selective maintenance. We can then test using data from new individuals whether selective maintenance predicts relapse to a

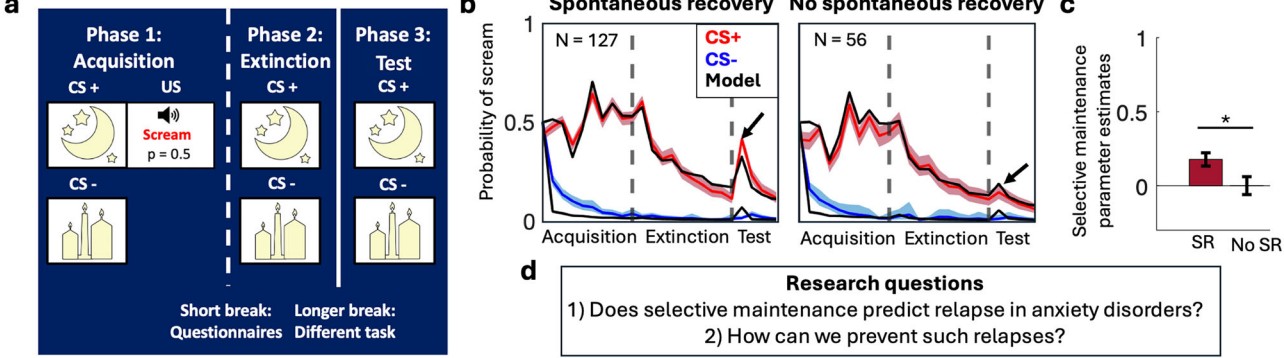

**Fig. 3 | Measuring learning tendencies that may underlie psychological interventions through a behavioral task. a** Task design. In acquisition, participants learned that one stimulus (CS + ) is followed by an aversive scream (US) on half the trials, and another stimulus (CS-) is not followed by a scream. Following a short break, in an extinction phase, participants saw both stimuli again without the scream, analogous to exposure therapy for anxiety. After a break of ~15 min, participants were shown the stimuli again to test for spontaneous recovery of fear of the scream. **b** Behavioral data (predictions of scream for the CS + (red), and CS- (blue)) reveal that some participants show spontaneous recovery of scream predictions at

the test phase (left) while others do not. In black are data simulated by the selective maintenance model using the best-fit parameters for each participant, illustrating that the model can account for these individual differences. Shading indicates 95% bootstrapped confidence intervals. Dashed lines indicate breaks. **c** Median estimates of the selective maintenance parameter were larger for participants who showed spontaneous recovery (SR; left) than those who did not (No SR; right), suggesting that selective maintenance might drive spontaneous recovery. **d** Examples of research questions that can be addressed with the combination of the task and the model. Figure adapted from Berwian et al.[15].

clinical useful degree. Second, we can use tasks (and/or computational models) to emulate (simulate) learning during different variations of psychological interventions and test for reduction of spontaneous recovery (see ref. 15 for an example).

Two points are crucial for testing such predictions: (1) the tasks must have very good psychometric properties in measuring individual differences[82], and (2) they need to be employed in longitudinal studies[83,84]. To obtain predictive validity, high task reliability and convergent validity need to be achieved first, as they set an upper bound for predictive validity. While many currently used tasks do not have sufficiently good psychometric properties for clinical usefulness, this issue has recently received more attention in the field of Computational Psychiatry, and solutions to improve psychometric properties and their implementations are emerging[82,84]. Test-retest reliability refers to consistent measurements by a task across time. The key to increasing test-retest reliability is decreasing noise in measurement of quantities of interest. This can be done through modeling choices such as hierarchical fitting of model parameters, joint fitting of different sessions and/or choices and reaction times and avoidance of difference scores[84]. It can also be achieved through task design choices such as including explicit practice trials to avoid early practice effects in task data, including breaks to minimize changes over time due to fatigue, including more trials to allow more precise estimates of quantities of interest, and other modifications that enhance the experimental effect and adjust the task difficulty to the population of interest[82]. Sufficient reliability has also been achieved for some tasks and models (e.g., refs. 85,86) indicating that while challenging, this is an obtainable goal. Task reliability is particularly important when using tasks to track changes due to interventions, e.g., in parameters such as reward or effort sensitivity[78].

Longitudinal studies are important because our hypotheses make predictions about how people with symptoms respond to specific treatments, not individual differences between people with and without a diagnosis or symptoms. We would like to stress that we believe that it is crucial to address questions relating to understanding and predicting treatment response, in order to help people seeking treatment and clinicians in their practice. This is because the question 'what treatment would be most helpful for this individual?' may be harder to answer (and more consequential for clients) than the question of 'what diagnosis does this individual have?'. In particular, for very heterogeneously and descriptively defined mental disorders such as depression, identifying subgroups with different longitudinal disease courses after treatment is likely to be clinically useful. Difficulties in recruiting participants with clinical diagnoses often impede the implementation of appropriate studies. Psychotherapy is also often not delivered in line with manuals[87], and even if manuals are used, they vary, are often tailored to the client and therapist, and not delivered exactly as described, making it difficult to tease apart individual variability from treatment variability as sources of differences in treatment outcomes. Finally, although we described exposure therapy and cognitive restructuring separately, these interventions are not always applied serially. For example, we described cognitive restructuring in line with recent manuals[57,88], but in his early manual of cognitive therapy for depression, Beck suggests "collaborative empiricism" – testing new thoughts and their predictions in behavioral experiments in the real world[89]. Such testing is sometimes included explicitly in the therapy alongside cognitive restructuring or might be done spontaneously by the client between sessions. Behavioral testing adds experiential, model-free learning to cognitive restructuring and likely supports the translation of model-based values to model-free ones, and thus would influence our predictions for treatment response. Internet-delivered CBT (iCBT) is one way to ensure that interventions are delivered in isolation and uniformly across study participants[62]. Hence, more manualized and structured therapies or iCBT might be useful to test predictions of learning theory for CBT, at least in a first instance.

### Extensions to other interventions and treatment effects

We believe that computational models of learning can be used to formalize all psychotherapy interventions that aim to achieve change. While it is beyond the scope of this manuscript to review all relevant theories and empirical evidence, we illustrate a few examples of applications of computational models to interventions beyond CBT. Transference, a psychodynamic intervention, refers to the idea that clients generalize from their experience with significant others to the therapist[90]. According to psychodynamic theories, clients create expectations that the therapist will behave similarly to those significant others. If the therapist recognizes this expectation, they can respond differently, for example, intentionally contrasting their response to the expected one. Ideally, this would help the client change their expectations about other relationships. We suggest that working with transference is based on model-free learning through experience with people in therapy. Thus, the underlying learning mechanisms may be similar to exposure therapy. As we described, according to the latent-cause framework, the client can update an existing latent cause or infer a new cause to predict how people will act towards them in the future. Which route is taken might depend on how strongly therapists counteract the clients' expectation, in line with the role of the magnitude of expectancy violations in updating old versus inferring new latent causes[42,91]. Crucially, formalizing the learning processes underlying psychotherapy interventions can help to illuminate similarities between interventions that are often considered different and separate due to their historical development in specific schools of psychotherapy.

Indeed, overall impaired learning can suggest that both CBT and psychodynamic therapy may not be effective methods for a given person, and perhaps pharmacological treatment should be the first line of action. Thus, parameters of formal learning models might be also useful to predict differential responses to psychotherapy versus pharmacotherapy more generally. Alternatively, as we suggested above, one can take advantage of hierarchical models that account for higher-level parameters such as prior beliefs about stochasticity of the world or reliability of experience that influence the rate or type of learning from interventions, and begin with interventions that change high-level parameters to allow better responsivity to later psychotherapy interventions.

Finally, computational models might also help to explain the strong effect of 'common factors' on treatment outcomes after different psychotherapy interventions[92]. One 'common factor' is the empirically well-supported finding that the alliance between the client and therapist is associated with treatment response[93,94]. The importance of this factor could be accounted for computationally by an increase in learning rates during therapy when the relationship is good and the client trusts the therapist. Formulated in Bayesian terminology, increased learning rates would be the result of more precision (less variance) around the new information received in therapy due to the trust towards the therapist.

### Conclusion

Advances in computational models of learning, together with a set of tasks used to measure parameters of the learning process, might help to explain and predict psychotherapy intervention effects. Of course, the predictions we outlined must be tested in longitudinal clinical studies. Training researchers to speak both the language of psychotherapy and of computational modeling is crucial for the advancement of this approach. Hopefully, this research can lead to better-informed assignment of people to the treatment most likely to be effective for them. When it comes to the integration of such assignments into the clinical setting, it is important to keep in mind that client wishes should always be kept paramount, as they strongly affect the client's motivation to learn and to do the work required for psychotherapy treatment to succeed. By laying out the relevant theories of learning and psychotherapies, we hope to move towards this goal and contribute to developing theory-driven 'Computational Psychotherapy.'

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

## Acknowledgements

We would like to thank Quentin Huys and the Computational Psychotherapy Discussion Group for helpful discussions around the topic of this paper. Isabel Berwian and Yael Niv's work on "Precision psychiatry for treatment selection in depression" is supported by Wellcome Leap as part of the Multi-Channel Psych Program. Wellcome Leap reviewed this Acknowledgement statement and the manuscript for intellectual property considerations.

## Author contributions

Isabel Berwian and Yael Niv developed the ideas and planned this paper. Isabel Berwian wrote the manuscript under the supervision of Yael Niv.

Peter Hitchcock and Sashank Pisupati provided useful contributions to interpretations of the clinical and computational modeling literature. Gila Schoen provided useful perspectives on CBT and its use in the clinic. All authors provided critical comments and read and approved the manuscript.

## Competing interests

The authors declare no competing interests.
