## [Transparent Peer Review file · Communications Psychology]

Using computational models of learning to advance cognitive behavioral therapy

Corresponding Author: Dr Isabel Berwian

This manuscript has been previously reviewed at another journal. This document only contains information relating to versions considered at Communications Psychology.

Version 0:

Decision Letter:

Dear Dr Berwian,

Your manuscript titled "Using computational models of learning to advance cognitive behavioral therapy" has now been seen by our reviewers, whose comments appear below. In light of their advice I am delighted to say that we are happy, in principle, to publish a suitably revised version in Communications Psychology.

We therefore invite you to revise your paper one last time to address the remaining concerns of our reviewers and a list of editorial requests. Please ensure to make use of Reviewer #1's constructive remarks to improve the overview of relevant literature. At the same time we ask that you edit your manuscript to comply with our format requirements and to maximise the accessibility and therefore the impact of your work.

EDITORIAL REQUESTS:

Please review our specific editorial comments and requests regarding your manuscript in the attached "Editorial Requests Table" and edited manuscript file. Please outline your response to each request in the right hand column. Please upload the completed table with your manuscript files as a Related Manuscript file.

SUBMISSION INFORMATION:

OPEN ACCESS:

At acceptance, you will be provided with instructions for completing the open access licence agreement on behalf of all authors. This grants us the necessary permissions to publish your paper.

*** TRANSPARENT PEER REVIEW:** Communications Psychology uses a transparent peer review system. On author request, confidential information and data can be removed from the published reviewer reports and rebuttal letters prior to publication. If you are concerned about the release of confidential data, please let us know specifically what information you would like to have removed. Please note that we cannot incorporate redactions for any other reasons.

Link Redacted

Best regards,

Jennifer Bellingtier

Jennifer Bellingtier, PhD
Senior Editor
Communications Psychology

REVIEWERS' EXPERTISE:

Reviewer #1 computational psychiatry
Reviewer #2 computational psychiatry, CBT

REVIEWERS' COMMENTS:

Reviewer #1 (Remarks to the Author):

This paper presents an interesting exploration of how computational models of learning, particularly reinforcement learning and latent-cause inference, can enhance our understanding of cognitive behavioral therapy, specifically exposure therapy and cognitive restructuring. The authors propose that these computational frameworks can offer a mechanistic explanation for individual differences in treatment responses, helping to predict therapy outcomes and aid interventions. While the theoretical mapping of learning models onto psychotherapy is insightful, I think the manuscript could be improved by providing more careful considerations surrounding the measurement of individual differences.

1. I appreciate the authors emphasizing the importance of addressing reliability issues in measuring individual differences using behavioral tasks. However, the current wording seems to downplay the enormity of this challenge and fails to mention that many currently used tasks and models do not have high test-retest reliability. In other words, there is a lot of work to be done on the reliability front before predictive validity even becomes a possibility. It would be good to discuss these issues in more detail. This paper could be a good resource to consult:

Karvelis, P., Paulus, M. P., & Diaconescu, A. O. (2023). Individual differences in computational psychiatry: A review of current challenges. *Neuroscience and Biobehavioral Reviews*, 148, 105137.
<https://doi.org/10.1016/j.neubiorev.2023.105137>

2. Your main argument is that understanding individual differences in these processes could inform treatment. However, it would help to explain in more detail how such individual differences would be determined.

a. Describe in more detail how tasks would be designed to study individual differences in these processes and how that would translate into treatment response prediction. It would help to add a diagram here. Perhaps you could draw inspiration from Figure 1 in

Gueguen, M. C., Schweitzer, E. M., & Konova, A. B. (2021). Computational theory-driven studies of reinforcement learning and decision-making in addiction: what have we learned? *Current Opinion in Behavioral Sciences*, 38, 40–48.
<https://doi.org/10.1016/j.cobeha.2020.08.007>

And figure 1 in

Patzelt, E. H., Hartley, C. A., & Gershman, S. J. (2018). Computational Phenotyping: Using Models to Understand Individual Differences in Personality, Development, and Mental Illness. *Personality Neuroscience*, 1.
<https://doi.org/10.1017/pen.2018.14>

b. In this context, you should elaborate on the assumption that fitting these models to performance in very simplistic scenarios would have predictive validity in the context of treatment response. This seems to be extremely optimistic given that current RL model parameters do not even show convergent validity among similar tasks:

Eckstein, M.K., Wilbrecht, L., Collins, A.G., 2021. What do reinforcement learning models measure? interpreting model parameters in cognition and neuroscience. *Curr. Opin. Behav. Sci.* 41, 128–137.

Eckstein, M.K., Master, S.L., Xia, L., Dahl, R.E., Wilbrecht, L., Collins, A.G., 2022. The interpretation of computational model parameters depends on the context. *Elife* 11, e75474.

I think that is a major obstacle to discuss and should not be glossed over. Convergent validity seems to be upstream predictive validity. Thus, if the former cannot be achieved, it would be naïve to expect parameter estimates to be predictive of anything outside of the context of the task itself. And if they are not predictive, then it becomes unclear what clinical utility the exercise of computational formalization can offer.

In the discussion you write: "High task reliability means that as long as the person has not changed fundamentally (e.g., through psychotherapy), they are assessed consistently by the task (e.g., similarly rank-ordered compared to other people; Zorowitz & Niv, 2023). Thus, it suggests that measured individual differences are due to differences in the mechanisms of interest and not momentary fatigue or other sources of noise (Hitchcock et al., 2022)." This is misleading. High reliability does not guarantee validity (ie, that you are measuring the mechanisms of interest).

Reviewer #2 (Remarks to the Author):

I would like to thank the authors for the excellent revision of their manuscript which benefitted both in content and structure. I have no further comments, but only one recommendation: please use the term "mental disorders" where appropriate (and not psychiatric conditions or psychiatric disorders).

Please find below the reviewers' comments in black, our response in blue and changes in the text in green.

REVIEWERS' COMMENTS:

Reviewer #1 (Remarks to the Author):

This paper presents an interesting exploration of how computational models of learning, particularly reinforcement learning and latent-cause inference, can enhance our understanding of cognitive behavioral therapy, specifically exposure therapy and cognitive restructuring. The authors propose that these computational frameworks can offer a mechanistic explanation for individual differences in treatment responses, helping to predict therapy outcomes and aid interventions. While the theoretical mapping of learning models onto psychotherapy is insightful, I think the manuscript could be improved by providing more careful considerations surrounding the measurement of individual differences.

Thank you for your positive assessment of our work and your helpful feedback. We agree that establishing reliable and valid measurements of the constructs is crucial for the success of the outline research approach. We now include discussion of such considerations and work on this question in our manuscript, as detailed below.

1. I appreciate the authors emphasizing the importance of addressing reliability issues in measuring individual differences using behavioral tasks. However, the current wording seems to downplay the enormity of this challenge and fails to mention that many currently used tasks and models do not have high test-retest reliability. In other words, there is a lot of work to be done on the reliability front before predictive validity even becomes a possibility. It would be good to discuss these issues in more detail. This paper could be a good resource to consult:

Karvelis, P., Paulus, M. P., & Diaconescu, A. O. (2023). Individual differences in computational psychiatry: A review of current challenges. *Neuroscience and Biobehavioral Reviews*, 148, 105137. <https://doi.org/10.1016/j.neubiorev.2023.105137>

We fully agree with the reviewer that the paucity of reliable tasks is an issue in the field. We found the paper by Karvelis et al. very insightful regarding both the issue in the field and potential solutions.

We have now included an overview of the problem and ways to address it in the 'Discussion' section:

Two points are crucial for testing the predictions: 1) the tasks must have very good psychometric properties in measuring individual differences (Zorowitz & Niv, 2023), and 2) they need to be employed in longitudinal studies (Hitchcock et al., 2022; Karvelis et al., 2023).

To obtain predictive validity, high task reliability and convergent validity need to be achieved first, as they set an upper bound for predictive validity. While many currently used tasks do not have sufficiently good psychometric properties for clinical usefulness, this issue has recently received more attention in the field of Computational Psychiatry, and solutions to improve psychometric properties and their implementations are emerging (Karvelis et al. 2023; Zorowitz & Niv, 2023). Test-retest

reliability refers to consistent measurements by a task across time. The key to increasing test-retest reliability is decreasing noise in measurement of quantities of interest. This can be done through modeling choices such as hierarchical fitting of model parameters, joint fitting of different sessions and/or choices and reaction times and avoidance of difference scores (Karvelis et al. 2023). It can also be achieved through task design choices such as including explicit practice trials to avoid early practice effects in task data, including breaks to minimize changes over time due to fatigue, including more trials to allow more precise estimates of quantities of interest, and other modifications that enhance the experimental effect and adjust the task difficulty to the population of interest (Zorowitz & Niv, 2023) . Sufficient reliability has also been achieved for some tasks and models (e.g., Waltman et al. 2022; Zorowitz et al. preprint), indicating that while challenging, this is an obtainable goal. Task reliability is particularly important when using tasks to track changes due to interventions, e.g., in parameters such as reward or effort sensitivity (e.g., Reiter et al., 2021).

2. Your main argument is that understanding individual differences in these processes could inform treatment. However, it would help to explain in more detail how such individual differences would be determined.

a. Describe in more detail how tasks would be designed to study individual differences in these processes and how that would translate into treatment response prediction. It would help to add a diagram here. Perhaps you could draw inspiration from Figure 1 in

Gueguen, M. C., Schweitzer, E. M., & Konova, A. B. (2021). Computational theory-driven studies of reinforcement learning and decision-making in addiction: what have we learned? *Current Opinion in Behavioral Sciences*, 38, 40–48.
<https://doi.org/10.1016/j.cobeha.2020.08.007>

And figure 1 in

Patzelt, E. H., Hartley, C. A., & Gershman, S. J. (2018). Computational Phenotyping: Using Models to Understand Individual Differences in Personality, Development, and Mental Illness. *Personality Neuroscience*, 1. <https://doi.org/10.1017/pen.2018.14>

Thank you for this helpful suggestion. We agree that a more detailed description of an implementation of the proposed research approach would be beneficial to the reader. We have now included in Figure 3 and the ‘Discussion’ section examples of a task, modeling the task data to infer underlying mechanisms, and clinical questions that these mechanisms may be able to address:

We so far provided a theoretical mapping between computational models of learning and psychotherapy research. To test the hypotheses generated with this approach, we need to measure learning tendencies and abilities in individuals. In Fig. 3, we illustrate this approach using a behavioral task that measures learning during extinction training, which emulates exposure therapy. We fit a version of the latent cause model introduced above to behavior elicited by the task in order to quantify underlying learning tendencies (for details on the technical steps involved in fitting models to behavioral data, see Wilson et al. (2019)). Our results suggested that some individuals show spontaneous recovery of fear with time, while other do not and that this difference could be related to differences in selective maintenance of adverse events in the model (Berwian et al., 2024). This task, and its related computational model allows us to ask new questions, e.g., whether we can predict who will be at higher risk of spontaneous recovery (relapse) of fear after exposure therapy for anxiety, and

whether the model's insights into the mechanism of spontaneous recovery can help to modify treatments so as to prevent relapse from occurring.

Fig. 3: Measuring learning tendencies that may underlie psychological interventions through a behavioral task

a) Task design. In acquisition, participants learned that one stimulus (CS+) is followed by an aversive scream (US) on half the trials, and another stimulus (CS-) is not followed by a scream. Following a short break, in an extinction phase, participants saw both stimuli again without the scream, analogous to exposure therapy for anxiety. After a break of ~15 min, participants were shown the stimuli again to test for spontaneous recovery of fear of the scream. b) Behavioral data (predictions of scream for the CS+ (red), and CS- (blue)) reveal that some participants show spontaneous recovery of scream predictions at the test phase (left) while others do not. In black are data simulated by the selective maintenance model using the best-fit parameters for each participant, illustrating that the model can account for these individual differences. Shading indicates 95% bootstrapped confidence intervals. Dashed lines indicate breaks. c) Median estimates of the selective maintenance parameter were larger for participants who showed spontaneous recovery (SR; left) than those who did not (No SR; right), suggesting that selective maintenance might drive spontaneous recovery. d) Examples of research questions that can be addressed with the combination of the task and the model. Figure adapted from Berwian et al. (2024).

As shown in the example in Fig. 3, tasks, models, and parameter estimates can be used to address such questions in two ways: First, we can train a machine-learning algorithm to predict relapse using model-derived parameter estimates of selective maintenance. We can then test using data from new individuals whether selective maintenance predicts relapse to a clinically useful degree. Second, we can use tasks (and/or computational models) to emulate (simulate) learning during different variations of psychological interventions and test for reduction of spontaneous recovery (see Berwian et al., 2024, for an example).

b. In this context, you should elaborate on the assumption that fitting these models to performance in very simplistic scenarios would have predictive validity in the context of treatment response. This seems to be extremely optimistic given that current RL model parameters do not even show convergent validity among similar tasks:

Eckstein, M.K., Wilbrecht, L., Collins, A.G., 2021. What do reinforcement learning models measure? interpreting model parameters in cognition and neuroscience. *Curr. Opin. Behav. Sci.* 41, 128–137.

Eckstein, M.K., Master, S.L., Xia, L., Dahl, R.E., Wilbrecht, L., Collins, A.G., 2022. The interpretation of computational model parameters depends on the context. *Elife* 11, e75474.

I think that is a major obstacle to discuss and should not be glossed over. Convergent validity seems to be upstream predictive validity. Thus, if the former cannot be achieved, it would be naïve to expect parameter estimates to be predictive of anything outside of the context of the task itself. And if they are not predictive, then it becomes unclear what clinical utility the exercise of computational formalization can offer.

Thank you for this comment. We agree that convergent validity of tasks measuring the same construct on the same range of the spectrum is a necessary pre-requisite for predictive validity. However, as Eckstein et al. (2022) noted as well, learning parameters are normatively influenced by specifics of the task (such as reward stochasticity or task volatility), and thus we may not expect their convergent validity unless tasks are matched for such contexts. Thus, we need to carefully tease apart in which cases we should expect convergent validity or which task aspects might prevent it. We have shown in some published work (e.g. Berwian et al. 2020) and in work that we are currently preparing for publication, that task behavior can predict outcomes after intervention (e.g. after behavioral activation) above and beyond clinical and demographic variables out-of-sample. Hence, given this proof-of-concept, we are indeed optimistic that the proposed research approach may yield clinically useful results.

In the discussion you write: “High task reliability means that as long as the person has not changed fundamentally (e.g., through psychotherapy), they are assessed consistently by the task (e.g., similarly rank-ordered compared to other people; Zorowitz & Niv, 2023). Thus, it suggests that measured individual differences are due to differences in the mechanisms of interest and not momentary fatigue or other sources of noise (Hitchcock et al., 2022).” This is misleading. High reliability does not guarantee validity (ie, that you are measuring the mechanisms of interest).

We agree that the wording is misleading. We have removed the second sentence and replaced it with a discussion of achieving relevant validity for tasks for the questions they aim to address (see our changes in the text copied above).

Reviewer #2 (Remarks to the Author):

I would like to thank the authors for the excellent revision of their manuscript which benefitted both in content and structure. I have no further comments, but only one recommendation: please use the term "mental disorders" where appropriate (and not psychiatric conditions or psychiatric disorders).

We want to thank the reviewer for the positive feedback. Throughout the text, we have updated ‘psychiatric disorders’ to ‘mental health conditions.’